# A systemic cell cycle block impacts stage-specific histone modification profiles during *Xenopus* embryogenesis

**Daniil Pokrovsky** , **Ignasi Forné** , **Tobias Straub** , **Axel Imhof** , **Ralph A. W. Rupp** *

Ludwig-Maximilians-Universität München, Planegg-Martinsried, Germany

* ralph.rupp@bmc.med.lmu.de

**Data Availability Statement:** The authors confirm that all data underlying the findings are fully available without restriction. The mass spectrometry proteomics data have been deposited

## Abstract

Forming an embryo from a zygote poses an apparent conflict for epigenetic regulation. On the one hand, the de novo induction of cell fate identities requires the establishment and subsequent maintenance of epigenetic information to harness developmental gene expression. On the other hand, the embryo depends on cell proliferation, and every round of DNA replication dilutes preexisting histone modifications by incorporation of new unmodified histones into chromatin. Here, we investigated the possible relationship between the propagation of epigenetic information and the developmental cell proliferation during *Xenopus* embryogenesis. We systemically inhibited cell proliferation during the G1/S transition in gastrula embryos and followed their development until the tadpole stage. Comparing wild-type and cell cycle–arrested embryos, we show that the inhibition of cell proliferation is principally compatible with embryo survival and cellular differentiation. In parallel, we quantified by mass spectrometry the abundance of a large set of histone modification states, which reflects the developmental maturation of the embryonic epigenome. The arrested embryos developed abnormal stage-specific histone modification profiles (HMPs), in which transcriptionally repressive histone marks were overrepresented. Embryos released from the cell cycle block during neurulation reverted toward normality on morphological, molecular, and epigenetic levels. These results suggest that the cell cycle block by HUA alters stage-specific HMPs. We propose that this influence is strong enough to control developmental decisions, specifically in cell populations that switch between resting and proliferating states such as stem cells.

## Introduction

Different cell types are distinguished by specific chromatin states, which help to target DNA-binding factors to specific genomic regions. Covalent posttranslational modifications of histones (PTMs) contribute to these states in a pivotal manner [1]. The most abundant and functionally studied histone modifications are methylation (me), acetylation (ac), and phosphorylation (Ph), although many other modifications have been reported [2]. Transcriptionally active chromatin domains are characterized by a distinct array of histone marks.

to the ProteomeXchange Consortium via the PRIDE partner repository with the dataset identifier PXD021803.

**Funding:** This work was funded by the Deutsche Forschungsgemeinschaft (DFG, German Research Foundation) – SFB 1064. Project numbers: A12 to RR, DP; Z03 to AI, IF; Z04 to TS. The funders had no role in study design, data collection and analysis, decision to publish, or preparation of the manuscript.

**Competing interests:** The authors have declared that no competing interests exist.

**Abbreviations:** ac, acetylation; CNS, central nervous system; DMSO, dimethyl sulfoxide; ES, embryonic stem; HMP, histone modification profile; HUA, hydroxyurea and aphidicolin; HUAwo, HUA washout; me, methylation; MZT, maternal to zygotic transition; PCA, principal component analysis; Ph, phosphorylation; PRM, parallel reaction monitoring; PTM, posttranslational modification of histone; qRT/PCR, quantitative reverse transcription polymerase chain reaction; XIC, extracted ion chromatogram.

H3K27ac and H3K4me1 are associated with active enhancers [3], while high levels of H3K4me2/3 are found at the promoters of active genes [4]. Transcribed gene bodies are enriched in H3 and H4 acetylation [5], H3K36me3 [6], and H3K79me3 [7]. As shown by TALEN-mediated gene knockout of the histone-methyltransferase DOT1L in *Xenopus tropicalis*, H3K79me3 provides a notable example for a histone PTM that is largely dispensable during *Xenopus* embryogenesis but becomes essential in premetamorphic tadpoles [8]. In contrast, methylation of lysine residues 9 and 27 of H3 are hallmarks of repressive chromatin at silent gene loci [9,10]. H3K27me3 is associated with the formation of facultative heterochromatin, whereas H3K9me2/3 has important roles in the formation of constitutive heterochromatin. Methyl marks on these 2 lysines take part in regulating gene expression during development [11,12]. Altogether, histone PTMs ensure genomic integrity and control both adaptive and stable transcription modes to accommodate cell differentiation and physiological needs during development.

Histone modifications convey important information to early developmental programs in mammals [13]. The regulation of epigenetic plasticity in mammals has been difficult to address in vivo so far, and our knowledge is largely derived from in vitro differentiation of pluripotent embryonic stem (ES) cells [14–16]. The mechanisms, by which inductive signals shape the epigenome during ES cell differentiation, are still poorly understood. *Xenopus laevis* is a non-mammalian vertebrate model organism that allows to investigate in vivo the transition from pluripotency to committed cell states.

The epigenetic landscape of *Xenopus* embryos develops largely from an unprogrammed state, although maternally encoded proteins and epigenetic marks influence early transcriptional programs [17,18]. After maternal to zygotic transition (MZT) embryonic chromatin is considered permissive and receptive to inductive signals [19]. Hierarchical clustering of histone PTMs has identified histone modification landscapes, which distinguish different developmental stages. We have termed these patterns as stage-specific histone modification profiles (HMPs). They derive from quantitative fluctuations of histone PTMs, occurring during key developmental events such as germ layer formation, cell fate commitment, and organogenesis. After the blastula stage, HMPs gradually shift toward repressive histone PTMs [20], in agreement with chromatin results from differentiating mammalian ES cells [21].

We have interpreted these stage-specific HMPs as part of a developmental program, which acts in parallel to the unfolding genetic pathways. Such a program might be controlled through mechanisms, which modulate either the expression or the kinetic activity of histone modifying enzymes. Cell proliferation could represent an additional mechanism, since histone incorporation during S phase dilutes preexisting histone modifications and both cell cycle length and proliferative status of embryonic cells are subject to dramatic changes. At early stages of *Xenopus* development (similar to zebra danio and *Drosophila*), the cell cycle is driven by an autonomous biochemical oscillator, which is unaffected by developmental signals or checkpoints [22–24]. Cleavage divisions are extremely rapid (from 10 to 30 minutes) and lack gap phases. After MBT, gap phases appear, the cell cycle lengthens, and becomes asynchronous [25]. Subsequently, an increasing proportion of cells exits the cell cycle during differentiation. Indeed, the first postmitotic cells appear already at the early neurula stage [26]. They are thought to be in G0 phase; however, it was recently shown that quiescent cells can also be found in G2 [27]. This developmental regulation of cell division outside of the G1 phase allows to quickly reinitiate cell divisions later in development. The findings, that in both *Xenopus* and *Drosophila* neurogenesis periods of intense proliferation are interrupted by phases of quiescence, support this notion of the cell cycle being used in a regulatory manner [28,29]. These changes in the cell cycle are accompanied by changes in local chromatin structure. Important histone modifications such as trimethylated H3K27 or H4K20 are not present at significant levels prior to

zygotic genome activation, but greatly increase after the MBT, when large-scale changes in chromatin modifications happen [30,31]. Together, these data indicate that major changes in cell cycle regulation and the appearance of stage-specific HMPs occur in a developmentally coordinated manner.

Cell division can be blocked in *Xenopus* from gastrulation onward by incubating embryos with DNA synthesis inhibitors. Rollins and Andrews (1991) used aphidicolin to demonstrate that morphogenesis and regulation of gene expression are independent from DNA replication [32]. Harris and Hartenstein used a combination of hydroxyurea and aphidicolin (HUA) to reveal that cell type determination and differentiation in embryonic central nervous system (CNS) are independent from cell division [33].

Here, we demonstrate that *Xenopus* embryos, arrested systemically during the G1/S transition during gastrulation (NF10.5), develop largely normal and in synchrony with control siblings until the tadpole stage (NF32). However, cell cycle–arrested embryos display an altered histone modification landscape, in which specific modifications accumulate both precociously and in excess. If arrested embryos are released back into proliferation, the chromatin landscape reverts toward normal. Thus, our data show that stage-specific HMPs are sensitive to HUA-induced replication inhibition.

## Results

### HUA treatment: Experimental design

Early *Xenopus* embryos rely on the consumption of maternally supplied molecules, which restricts the available repertoire of cell cycle manipulations. We have chosen a systemic cell cycle arrest, achieved by a combination of HUA (from here on called HUA condition), because, together, they block S phase more rapidly and more completely than each molecule alone [33]. When applied to cells in culture, both drugs effectively block DNA replication and consequently cell division, without obvious side effects on cell viability or differentiation capacity [34,35]. This effectively leads to a cell cycle arrest at the G1/S transition. Harris and Hartenstein have used these inhibitors to demonstrate that cell division is dispensable for neural induction, neuronal differentiation, and for neural tube formation [33]. We have confirmed their finding that, whereas HUA treatment before gastrulation (i.e., at NF9) leads to aborted development, it is principally compatible with embryonic development until tadpole stages when started shortly after gastrulation onset (NF10.5) (for more information, see Embryos handling and HUA treatment in Materials and methods section and S1A Fig). In this work, we have assessed the effects of the systemic cell cycle arrest by 2 types of experiments (Fig 1). Experiment type A ("permanent arrest") implies a continuous HUA or Mock treatment on sibling embryos from NF10.5 until NF32. Mock treatment represents incubation in 2% dimethyl sulfoxide (DMSO), which is the solvent for aphidicolin. Experiment type B ("transient arrest") consists of continuous HUA and Mock treatments plus an additional HUA washout (HUAwo) condition. In the latter case, embryos were temporarily incubated with HUA from early (NF10.5) to late (NF13) gastrula stages, and then returned to Mock solution. Experiment type B tests the reversibility of the cell cycle arrest and its developmental consequences.

For all experiments, embryo siblings were collected at 4 developmental stages: NF13 (early neurula), NF18 (late neurula), NF25 (tailbud), and NF32 (tadpole) (Fig 1). By the early neurula stage, germ layers and body axes have been determined. The embryonic patterning increases the cellular diversity of the embryos during neurulation. At the tailbud stage, the first differentiated tissues such as skeletal muscle and the mucociliary epithelium of the larval skin have been formed. The end point of the analysis (NF32) is after the phylotypic stage of *Xenopus*

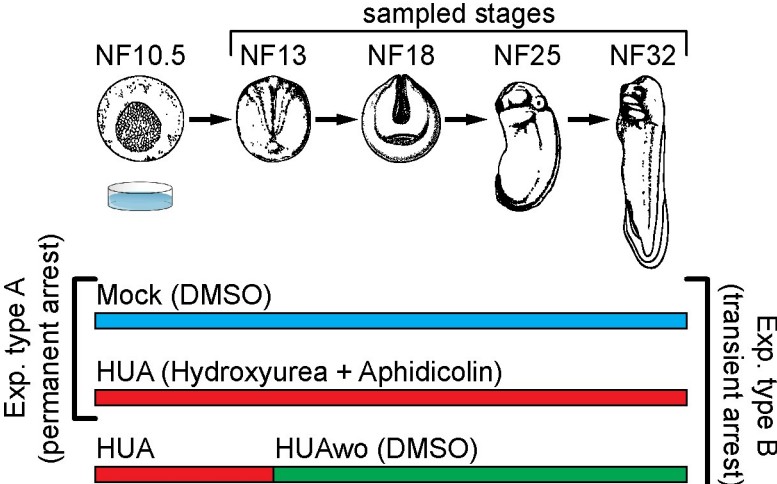

**Fig 1. Experimental design.** Top part: a schematic representation of *Xenopus laevis* embryonic development. Developmental stages (NF) according to Nieuwkoop and Faber (1994). Stages used for mass spectrometry of histone modifications and embryonic analyses are characterized by the following features: NF13—late gastrula, germ layer specified; NF18—neurula, germ layer patterning and differentiation; NF25—tailbud stage, organogenesis; NF32—early tadpole, body plan established. Bottom part: We perform 2 types of experiments: Experiment type A (G1/S block)—embryos are split in 2 groups, which from NF10.5 on are continuously incubated in HUA solution or control solution (DMSO as carrier). Experiment type B (transient arrest)—embryos are split in 3 groups: continuous Mock, continuous HUA, and transient HUA. In the last group, HUA solution is replaced at NF13 with DMSO solution. DMSO, dimethyl sulfoxide; HUA, hydroxyurea and aphidicolin; HUAwo, HUA washout.

(NF28 to NF31), when evolutionary conserved gene expression programs have established the vertebrate body plan [36].

## HUA effects on embryo viability and cell proliferation

At the first sampling stage (NF13), there was no difference in viability between proliferating controls and HUA-arrested embryos (S1A Fig). As reported before [33], HUA treatment reduced embryonic viability after NF18, such that the survival rate was about half of that of Mock embryos at NF25 and NF32. Staining for activated Caspase 3 indicated that both Mock- and HUA-treated embryos contained apoptotic cells in a variable, but comparable extent (S2 Fig), consistent with the results of an independent study using HUA in *Xenopus* [37]. Depending on individual egg batches, arrested embryos survived up to stage NF41.

To test the efficacy of the HUA incubation, we stained embryos for the histone H3 Serine 10 phospho mark (Fig 2), which accumulates in M phase cells [38]. We determined the number of H3S10Ph-positive cells from the entire surface of the embryo with ImageJ (Fiji). Already at NF13, HUA embryos contained more than 8-fold less mitotic cells than Mock embryos, a difference that further increased to 17-fold by NF32. We conclude that the cell cycle is efficiently blocked by HUA treatment.

## HUA effects on morphogenesis

The cell cycle–arrested embryos completed gastrulation in synchrony with Mock controls (Fig 3A). From midneurula onward (stage NF18/19), the closure of the neural tube was delayed, consistent with previous findings [33] (S1B Fig). At later stages, HUA embryos lacked a postanal tail, melanocytes, and the eye anlage remained rudimentary. Note that the size of arrested and Mock-treated embryos was almost the same until stage NF25 (Fig 3A). Since holoblastic cleaving *Xenopus* embryos rely completely on maternal storage, cell proliferation subdivides

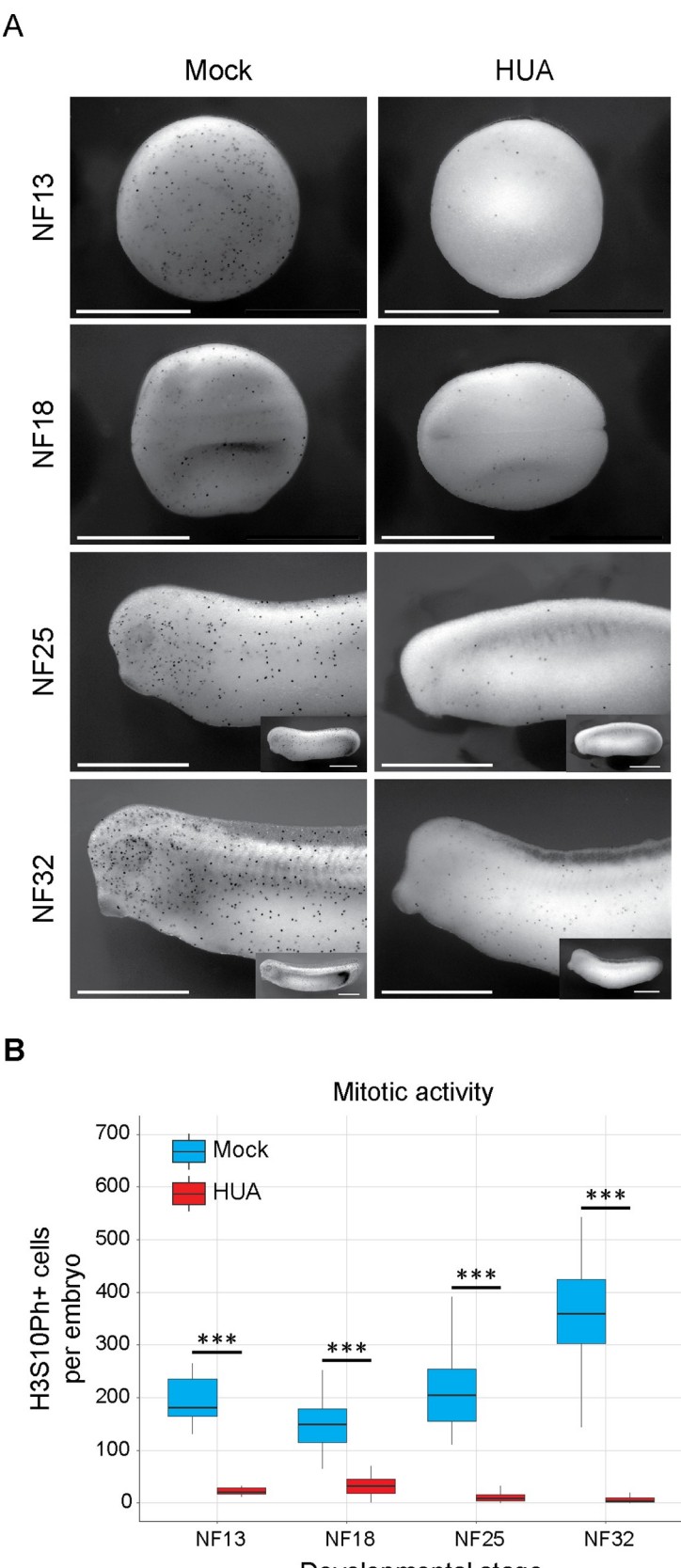

**Fig 2. Continuous HUA treatment inhibits mitotic activity from gastrula to tadpole stages.** (A) ICC for the mitotic histone mark H3S10Ph. Black dots represent mitotic cells. Elongated, older embryos are recorded as anterior halves, i.e., at the same magnification as younger stages, and in whole mount views as inserts. Scale bars: 1 mm. (B) Abundance of mitotic cells in Mock- and HUA-treated embryos. Box plots based on H3S10Ph-positive cells present on the recorded surface of embryos ($n = 3$ biological replicates/condition; Student $t$ test [unpaired, two-tailed]; ∗∗∗ $p < 0.001$). The individual quantitative observations can be found in S1 Data. HUA, hydroxyurea and aphidicolin; ICC, immunocytochemical staining.

the zygote into smaller and smaller cell progeny, while maintaining the maternally derived mass. Consistent with the observed mitotic arrest, we confirmed that HUA embryos consisted of bigger cells at the neurula stage (S1B Fig). This observation is consistent with the previous finding, showing that blocking cell division with HUA stops the decrease in cell size during embryonic development [39]. Confocal imaging indicates that epidermal cells of HUA embryos were clearly spread out over a larger area than proliferating cells of control embryos, and their nuclei appear enlarged (Fig 3B). H3S10Ph-positive cells were detected in Mock controls, but not in HUA embryos (S1 and S2 Videos). Overall, these observations document a remarkable compensation on the cellular level, which enables HUA embryos to undergo morphogenesis in apparent synchrony with control siblings.

## Differentiation of cell cycle–arrested embryos

Cell proliferation and differentiation are, in most cases, mutually exclusive processes [40]. To investigate the developmental relationship of Mock- and HUA-treated embryos, we investigated the tissue-specific expression of a panel of marker genes (Fig 3C and S1 Table). Whole mount RNA in situ hybridization revealed that most of these markers (17 out of 19) were expressed at the correct site, although in smaller domains. Only 2 genes were not transcribed in the majority of HUA-treated embryos. *Xbra* and *foxd5* are normally expressed in the tail, which HUA embryos fail to form. To rule out that the absence of these mRNAs in cell cycle–arrested embryos was due to a developmental delay, we compared by quantitative reverse transcription polymerase chain reaction (qRT/PCR) the timing and relative levels of gene transcription between Mock- and HUA-treated embryos (S1C Fig). We examined 6 genes, which become induced at different time points—*pax6* and *actc1* at gastrulation, *twist* and *myt1* during neurulation, and *tnni3* and *fabp2* in tadpoles—and normalized the relative mRNA levels to the housekeeping gene *odc*. A significant difference in gene expression between Mock and HUA embryos was found only for *pax6*, which had disappeared by stage NF32. The mRNA levels of the other 5 genes, including the late-induced *fabp2* gene, were proportionate between the 2 conditions. This suggests that genes were activated around the proper time, arguing against a general delay in the development of HUA embryos.

Although HUA-treated tadpoles were morphologically impaired, they responded to touch stimulation by muscle twitching, and at least some of them were capable of mounting a flight response (S3 and S4 Videos). This burst of swimming activity involves a physiological connection between sensory neurons, the CNS, and the body wall musculature [41].

The morphological and molecular analyses demonstrated that embryonic development proceeds in the absence of cell proliferation, although the anlagen of some organs like tail, fin, and eyes were compromised. These findings are in agreement with and extend previous observations on HUA-treated embryos [33]. We conclude that blocking the cell cycle from gastrula stages onward is compatible with largely normal and apparently synchronous differentiation of the embryo at least until tadpole stage (NF32).

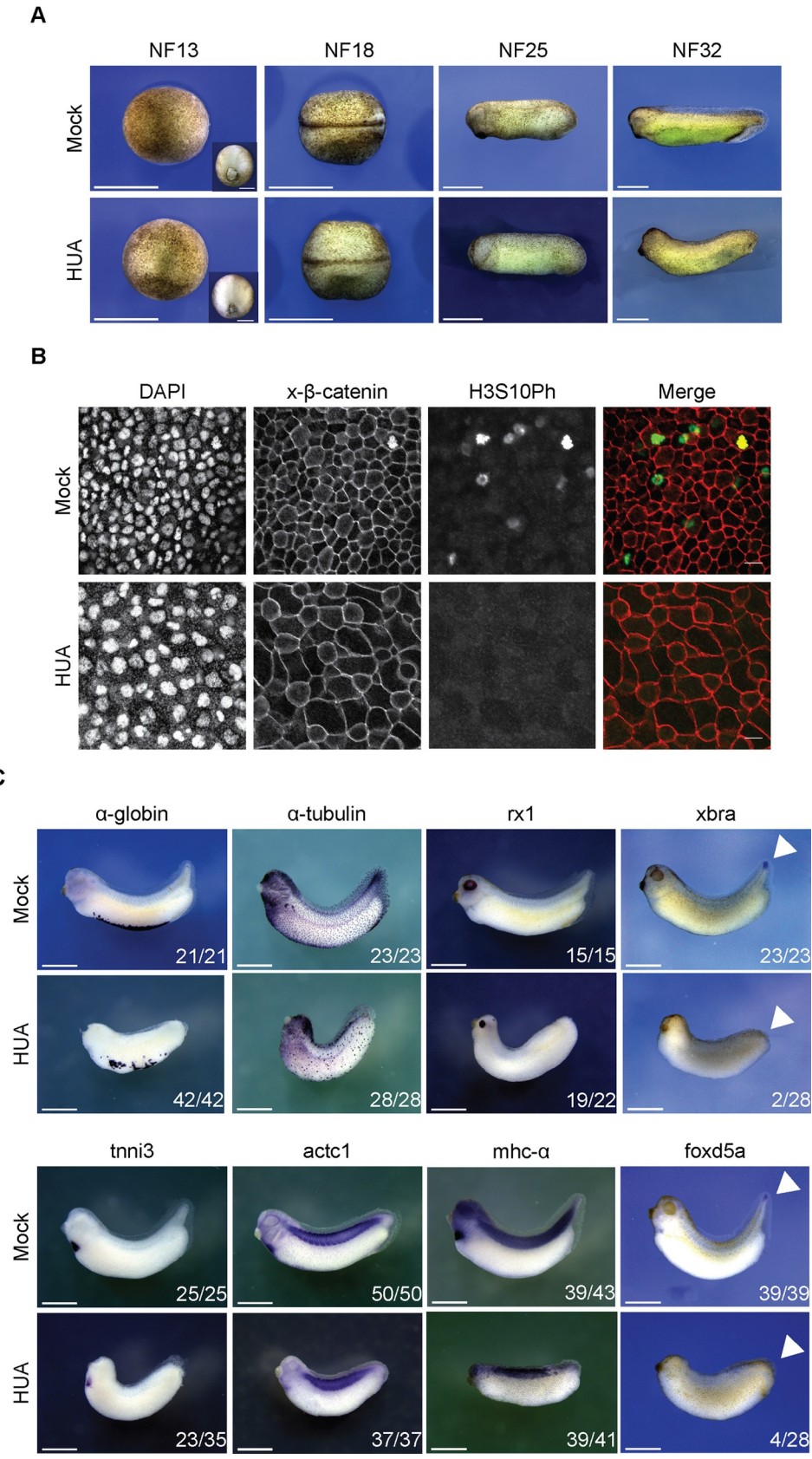

**Fig 3. Morphological development of HUA-arrested embryos.** (A) During gastrulation, Mock and HUA embryos are indistinguishable. At NF18, the latter show a delay in neural tube closure. More severe malformations are detectable at stages NF25 and NF32, most notably reduced eye formation and absence of tail bud. Scale bars: 1 mm. (B) Cell size at the tailbud stage. Flattened Z-stack images show fields from embryonic skin at constant magnification (scale bars: 20 μm). Immunofluorescence detects cell borders (beta-catenin), nuclei (DAPI), and mitotic cells (H3S10Ph). GIFs from Z-stacks are presented in the Supporting information (S1 and S2 Videos). (C) Whole mount RNA in situ hybridization for indicated marker genes. Images are representative for the majority of Mock- or HUA-treated embryos from 3 biological experiments. While mRNAs of α-tubulin and rx1 (skin, neuronal), α-globin, tnni3, actc1, and mhc-α (mesodermal) are detected in both conditions, xbra and foxd5a (tail blastema) are absent in HUA embryos. Numbers indicate embryos positive for the marker over the total number of analyzed embryos. Scale bars: 1 mm. HUA, hydroxyurea and aphidicolin.

## Stage-specific histone modification profiles in HUA embryos

To investigate the impact of the cell cycle arrest on embryonic chromatin, we compared the posttranslational HMPs in proliferating and HUA-arrested embryos by quantitative mass spectrometry (for more information, see Mass spectrometry analysis with scheduled PRM method and Histone PTM quantification in Materials and methods section). In total, this study quantified 64 histone modification states across 4 developmental stages in 2 experimental conditions, each in 3 biological replicates. The combination of all modification states at a certain developmental time point constitutes the so-called stage-specific HMP [20]. To visualize differences in the HMPs between cell cycle–arrested and control embryonic chromatin, we performed hierarchical clustering analysis with all 24 samples to build a single, stage-specific heatmap (Fig 4A). This clustering was based on the absolute intensity values of endogenous histone peptides.

The first 3 levels of the dendrogram on the left axis of the heatmap revealed 5 clusters with the following features. Cluster 1 grouped histone PTMs, which undulate in control embryos, i.e., they were more abundant at stages NF13 and NF25 than at stages NF18 and 32. In cell cycle–arrested embryos, this pattern collapsed into a single high point at NF25. Cluster 2 included modification states, which gradually increased their abundance under both experimental conditions. Cluster 3 represented modifications, which were more abundant in HUA-arrested chromatin compared to control chromatin, whereas in clusters 4 and 5, highly abundant modification states in Mock embryos were down-regulated in HUA samples.

Despite the fact that the cell cycle arrest alters the abundance of nearly every analyzed modification to varying extent, HMPs are clearly discernible for both conditions (Fig 4A). To obtain further insight into their difference, we performed principal component analysis (PCA) for the whole data set (S3A Fig). The 2 conditions are partly separating, particularly clear for late gastrula control and tadpole HUA samples. In general, chromatin features of younger HUA samples seem to cosegregate with those of older controls.

## HUA effects on individual histone marks

The global, absolute heatmap indicated that the HUA block resulted in a different developmental histone modification profile. To obtain a better understanding of how the histone modification states differed between the 2 conditions, we converted the absolute abundance of individual histone modifications into relative proportions for those modification states, which are located on a single tryptic histone peptide (S4 Fig and S5 Table). We then investigated the distribution of the relative histone PTM abundance over their fold-change between Mock and HUA (S3B Fig). Lower abundant modifications tend to have larger variability. Significantly different modifications (highlighted in red) are spread through the entire range of abundance but are more prevalent at the high abundance end. One-quarter of the analyzed histone PTMs is significantly different between Mock and HUA conditions (S3C Fig and S5 Table). We then

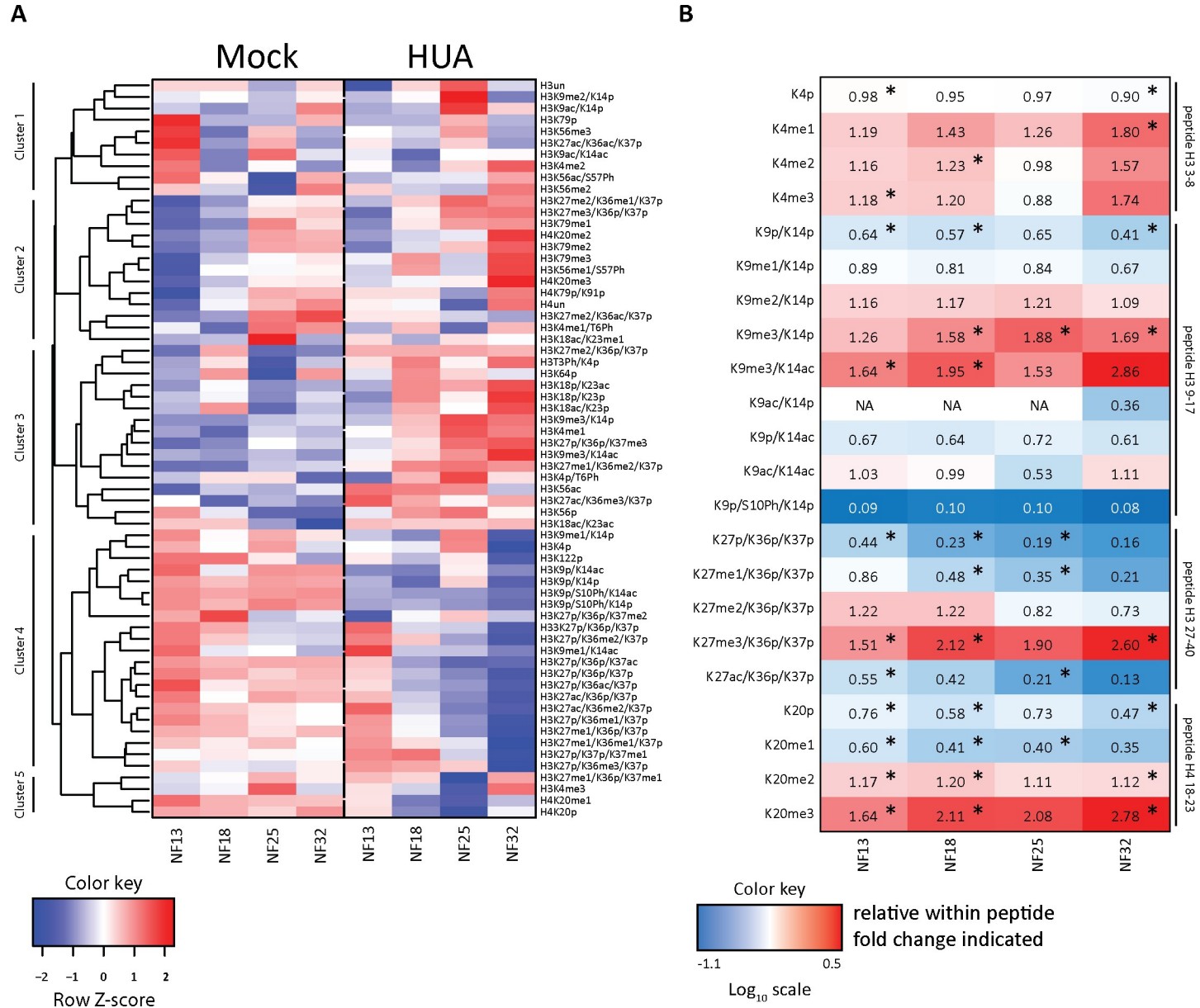

**Fig 4. Mitotic activity shapes stage-specific HMPs.** (A) Heatmap of the absolute histone modification states abundance, normalized to the corresponding R10 spiketides. Data in columns represent an average from 3 biological replicates/condition. Color key: row Z-score. The individual quantitative observations can be found in S2 Data. (B) Ratio heatmap of relative histone modifications abundance for selected histone marks. In the first step, relative distributions of the indicated modification states were calculated in percentages within each tryptic peptide (see S4 Fig). Then, HUA values were divided by Mock values to produce the relative change for each histone modification state between the 2 conditions. Color key is based on the Log10 scale. Numerical values in the cells indicate the fold-change. Values greater than 1.0 indicate an increase in HUA condition; values smaller than 1.0 indicate a higher abundance in Mock condition. NA, not applicable, when values in Mock were 0. Asterisks (*), adj.*p* < 0.1 (Benjamini–Hochberg procedure). "p," propionylated (naturally unmodified); "un," tryptic peptide, which has no modification state. The direct comparison of relative histone PTMs abundance is shown in S4 Fig, with *p*-values indicated in S5 Table. HMP, histone modification profile; HUA, hydroxyurea and aphidicolin; PTM, posttranslational modification of histone.

generated a second heatmap to display the fold-change of some selected histone PTMs between Mock and HUA conditions at a given stage (Fig 4B). These PTMs were selected based on their biological relevance for embryonic development and established impact on gene

activity. PTMs, whose fold-change between Mock and HUA conditions is statistically significant (adj.$p < 0.1$, Benjamini–Hochberg procedure), are marked in Fig 4B (and S5 Table).

The relative ratio heatmap supports the permanent nature of the G1/S cell cycle arrest by revealing on average a more than 10-fold reduction of the mitotic chromatin mark H3S10Ph. These data are in line with the assumption that HUA treatment affected not only superficial cell layers, but all embryonic tissues. Phosphorylation of Threonine 3 on histone H3 represents a second PTM that is mitotically enriched [42]. For unknown reasons, this modification shows a large variability between biological replicates in our data set, resulting in a very similar averaged abundance for Mock and HUA conditions in all 4 stages (S5 Table). Therefore, it cannot confirm our conclusions from the H3S10Ph mark.

Furthermore, the overall levels of H3K4me2/me3 were maintained in HUA embryos at all 4 stages, suggesting that the number of active and poised promoters can be maintained in non-proliferating embryos, consistent with the largely correct expression of cell-type specific markers (Figs 3C and 4B). Other modifications deviated between the 2 conditions. The embryonic epigenome of HUA-treated embryos contained reduced levels for some modifications, notably the acetylated states of K9 and K27 on histone H3. These modifications are thought to prevent the writing of repressive methylation marks on chromatin harboring *cis*-regulatory DNA elements [43,44], a prerequisite for de novo enhancer activation [45]. Beyond the H3S10Ph mark, the next largest differences in HUA-arrested chromatin were found for the trimethylated states of lysines 9 and 27 on H3, and lysine 20 on H4. These sites became globally elevated by more than 2.5-fold compared to the chromatin of control embryos. While higher levels of H3K27me3 and H4K20me2/3 were anticipated, since their levels are linked to cell cycle progression [46], the increase in H3K9me3 suggests that the HUA cell cycle arrest occurs partly in S phase [47].

Altogether, we demonstrated that incubating *Xenopus* embryos in HUA solution was an efficient way to systemically block cell proliferation. The HUA-induced cell cycle block had consequences on defined morphological aspects of development but also altered stage-specific HMPs. Most notably, this had an impact on specific histone modifications known to be involved in epigenetic memory and enhancer activity [45,48]. The chromatin differences between HUA and control embryos suggest that the cell cycle state—possibly cell proliferation as such—coordinates the developmental decoration of chromatin with covalent histone modifications.

## HUA effects on embryogenesis are reversible

To check whether the observed differences in morphology, gene expression, and histone modification landscape are reversible, we performed a transient cell cycle arrest (HUAwo) (Fig 1). Only 4 hours after returning to Mock condition (i.e., at midneurula stage—NF18), H3S10Ph-positive cell numbers were significantly increased compared to siblings maintained in HUA (Fig 5A and S5 Fig). At the endpoint of the experiment, mitotic cells were equally abundant in HUAwo and Mock conditions, indicating full recovery of the mitotic activity within 13 hours after inhibitor removal. In addition, the survival rate of HUAwo embryos increased, compared to continuously arrested embryos, and was similar to that of Mock embryos (S1A Fig).

Morphological hallmarks also recovered toward normality in HUAwo embryos (Fig 5B). Unlike continuously arrested embryos, the HUAwo group frequently developed postanal tails of nearly normal length, as well as a clearly distinguishable fin surrounding trunk and tail. The recovery of the tail structure was accompanied by the recurrence of xbra and foxd5a mRNAs at the tail tip of all HUAwo embryos (Fig 5B). These findings indicate that the major morphological defects of the HUA arrest are reversible, at least when the inhibitors are removed during neurulation.

**A**

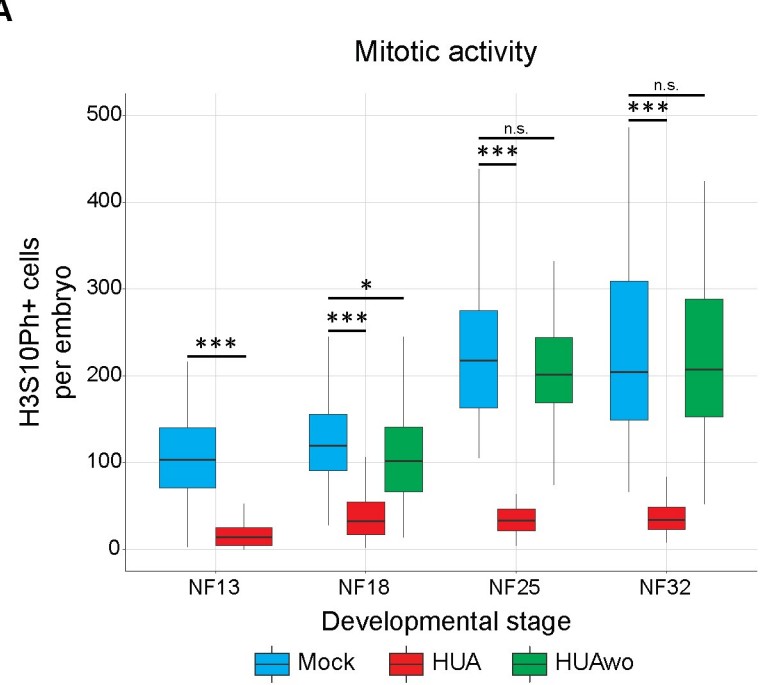

**B**

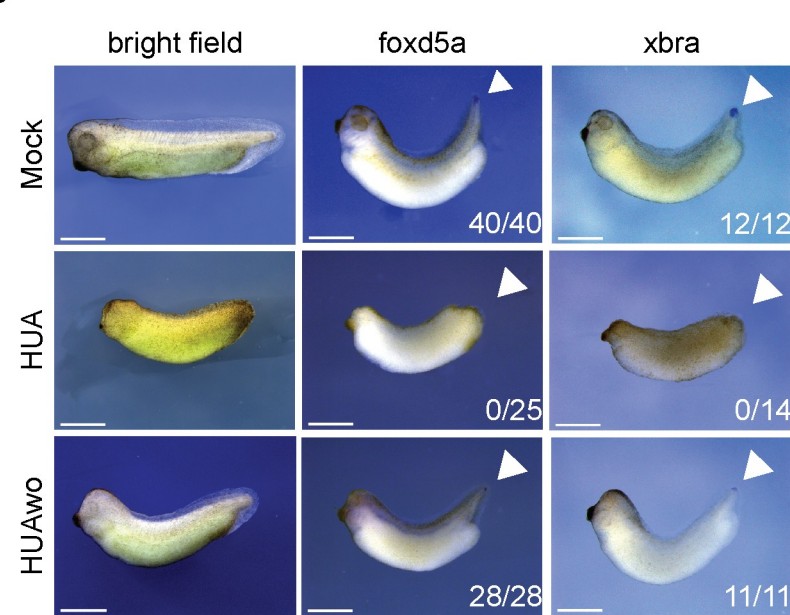

**Fig 5. The HUA effects on embryogenesis are reversible.** Embryos, which were transiently incubated in HUA solution and thus mitotically arrested, were returned to Mock solution at NF13. (A) Abundance of mitotic cells in Mock, HUA-treated, and HUAwo embryos. Box plots based on H3S10Ph-positive cells present on the recorded surface of embryos, displayed in S5 Fig (n ≥ 3 biological replicates/condition; Student *t* test [unpaired, two-tailed]; *** $p < 0.001$; * $p < 0.05$; n.s., not significant). The individual quantitative observations can be found in S3 Data. (B) Morphological and molecular features of embryos at the early tadpole stage. In contrast to continuous HUA-treated embryos, HUAwo embryos regain eye cups, fin, and tailbud structures. In addition, they express foxd5a and xbra mRNAs in the growth zone of the tailbud, comparable to Mock embryos. Numbers indicate embryos positive for the marker over the total number of analyzed embryos (n = 3 biological replicates/condition). Scale bars: 1 mm. HUA, hydroxyurea and aphidicolin; HUAwo, HUA washout.

Encouraged by the results, we next performed mass spectrometry analysis on sibling cohorts of Mock, permanent HUA, and HUAwo embryos at the tadpole stage (NF32). Experimental series type A and B include 2 similar conditions: Mock and permanent HUA. To compare results of the same conditions between Exp.A and Exp.B, we looked at the distribution of histone PTM fold-change at NF32 (S3D Fig). We did not observe a significant difference, supporting the robustness of the mass spectrometry measurements.

Next, we visualized the results of the Exp.B (HUAwo) in a heatmap, based on relative values (Fig 6A and S6 Table). As for the experiment type A, the fold-change of functionally well-annotated histone PTMs was calculated from their relative proportions (Fig 6B and S6 Fig). In total, almost three-quarters of the modifications were closer to the levels found in control embryos than to HUA chromatin, although in most cases, they do not reach wild-type abundance.

In detail, we found that H3S10Ph levels were 8-fold higher in transiently arrested (66% of Mock) versus constantly arrested (8% of Mock) embryos. The return to proliferation is also reflected in increased H4K20me1 levels. In proliferating cells, this modification reaches the maximum in G2 and M phases, when PR-Set7 monomethylates new histones incorporated during the previous S phase [49,50]. The observed doubling in H4K20me1 levels (Fig 6B and S6 Fig) indicates that upon inhibitor washout, cells can go through S phase and mitosis. Also, the H3K27me1 mark, which promotes the transcription of H3K36me3 decorated genes in ES cells [10], is recovering, as well as the abundance of acetyl marks on H3K9 and H3K27. The latter marks protect histone H3 from the repressive influence of trimethylation at these lysines, consistent with a transcriptionally more permissive state of chromatin. Indeed, levels of the repressive chromatin modifications H3K9me3, H3K27me2/3, and H4K20me3 were lower in the HUAwo condition, compared to permanently arrested embryos. Whether these changes are causally connected, i.e., acetylation marks replace methyl marks, or occur at independent chromatin regions, is not known. Nevertheless, the readjustments in the chromatin landscapes of embryos released from the G1/S block are in agreement with the observed morphological improvement.

Overall, these results document that HUA-arrested embryos can reinitiate cell proliferation, when the inhibitors are washed out at late gastrula. In contrast to embryos, treated continuously with the cell cycle inhibitors, the HUAwo embryos show a higher vitality with improved tissue formation and gene expression. These changes are accompanied by a partial restoration of their normal histone modification landscape. Taken together, the results of the inhibitor washout experiment support the hypothesis that cell proliferation has an impact on the development of stage-specific HMPs during early embryogenesis of *Xenopus*.

## Discussion

Establishing and maintaining the epigenetic information of covalent histone modifications faces a fundamental problem in proliferating cells, i.e., a 2-fold replicational dilution of preexisting histone marks during S phase needs to be matched by the biological kinetic rates of the enzymes decorating the chromatin landscape. While one might assume that evolution has ensured a robust balance between dilution and writing of histone PTMs, which adequately meets the physiological requirements of cells to acquire and maintain gene expression profiles and genome stability, recent work has indicated that this balance is delicate and could provide a resource for epigenetic regulation [51–54].

In this study, we have addressed the impact of HUA-induced cell cycle arrest on the histone modification landscape during *X. laevis* embryogenesis. We assume that the observed changes in stage-specific HMPs arise primarily in consequence to the cell cycle block. This assumption

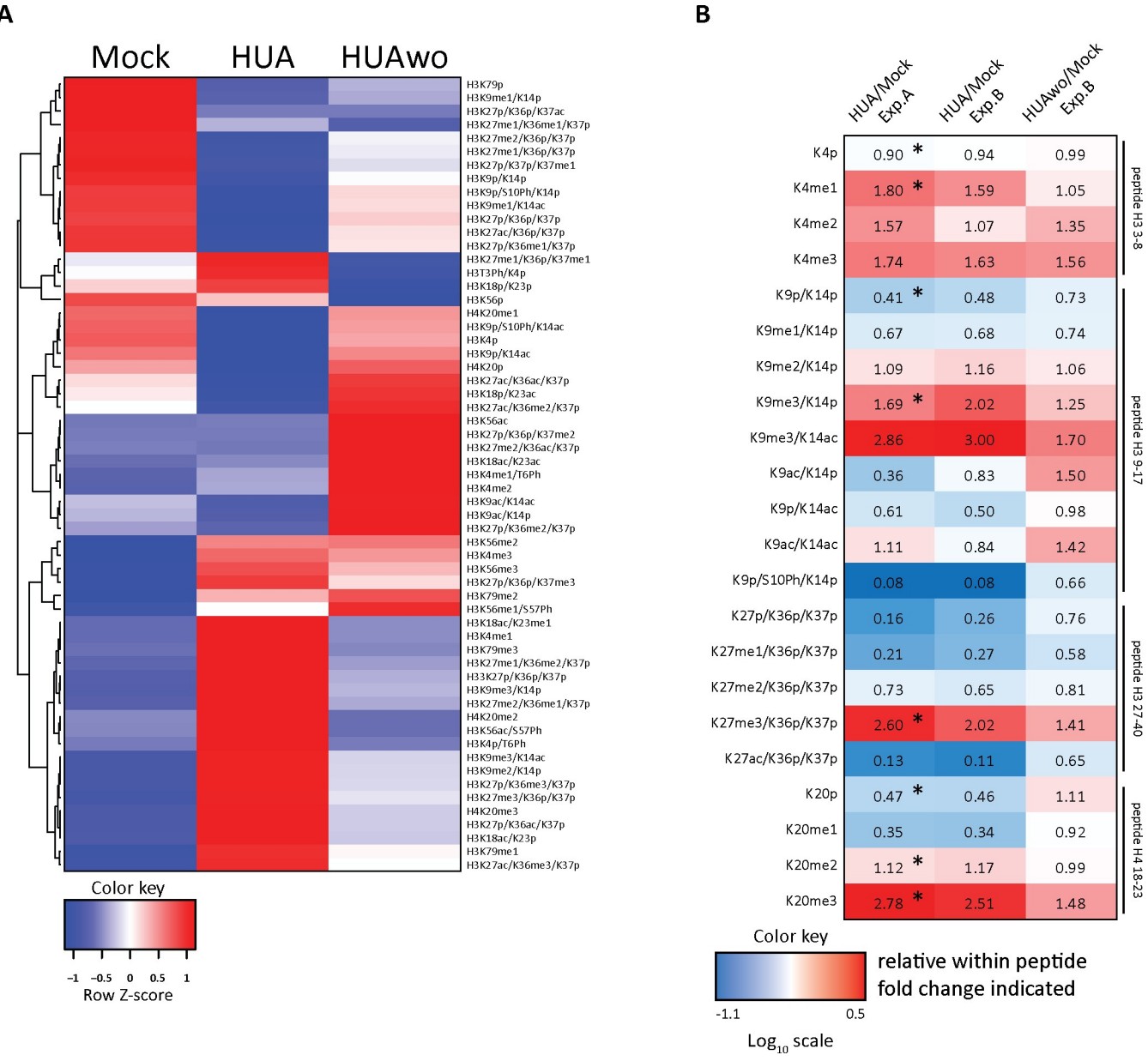

**Fig 6. HUA effects on stage-specific HMPs are reversible.** (A) Heatmap of the relative histone modification states abundance under the indicated conditions. As before, the relative distributions of the indicated modification states are calculated in percentages within each tryptic peptide. Data in columns are collected from early tadpole stage embryos NF32 (*n* = 1 biological replicate), independently from the data of experiment type A. Note: Relative differences between Mock and HUA samples of experiment type B are highly similar to the results of experiment type A (S6 Fig). Color key, row Z-score. The individual quantitative observations can be found in S4 Data. (B) Ratio heatmap of relative histone modification abundance for selected histone marks. First, the relative distributions of the indicated modification states are calculated as percentages within each tryptic peptide (S6 Fig). Then, the relative values are presented as ratios between sample pairs indicated on top, showing the relative change for each histone modification state between the 2 conditions. Color key is based on the Log10 scale. Numerical values in the cells indicate the fold-change. "p," propionylated (naturally unmodified). Values greater than 1.0 indicate an increase in HUA condition; values smaller than 1.0 indicate a higher abundance in Mock condition. NA, not applicable, when values in Mock were 0. The first 2 columns illustrate the similarity between experiment types A and B; the information in the third column shows the similarity between HUAwo and Mock samples. Asterisks (*), adj.*p* < 0.1 (Benjamini–Hochberg procedure) between indicated conditions. The direct comparison of relative histone PTMs abundance is shown in S6 Fig. HMP, histone modification profile; HUA, hydroxyurea and aphidicolin; HUAwo, HUA washout.

is based on multiple lines of independent evidence that testifies to the vitality of HUA-arrested embryos, such as intact embryonic patterning, correct spatiotemporal gene expression, survival to the tadpole stage, low apoptotic rate, and last but not least, the reversibility of morphological, molecular, and epigenetic phenotypes upon HUAwo. But, at this point, we cannot exclude that nonphysiological effects of the inhibitors contribute to the observed chromatin changes. Specifically, hydroxyurea causes both replication stress and DNA damage through its breakdown products [55,56]. However, since DNA damage is associated with significant removal and degradation of core histones from DNA [57,58], DNA damage and repair are unlikely to explain an increase in repressive PTMs, which are a hallmark of HUA chromatin. Finally, because *Xenopus* embryos develop by reductive proliferation, blocking the cell cycle leads to an increase in cell size. As recently shown, the ratio of surface area to volume of a cell determines the partitioning of IMPORTIN-a into cytosolic and plasma membrane–associated fractions, which scales intracellular structures, including nuclear size [59]. The relative availability of IMPORTIN-a for cytoplasmic nuclear transport could influence the nuclear concentration of histone modifying enzymes, or factors regulating them, and thereby alter the histone PTM abundance. In conclusion, both physiological and nonphysiological mechanisms may contribute to alterations in histone PTMs upon HUA treatment of embryos. In light of published evidence that S phase dilution influences the propagation and maintenance of histone marks in cell lines [46,60], we consider the cell cycle block as a primary driver for the observed changes in histone PTM abundance.

*Xenopus* embryos represent a model system, in which the cell cycle naturally undergoes massive changes. In this system, modulations in cell cycle phases, cell cycle length, and differentiation-coupled withdrawal from cell cycle are tightly coordinated with developmental programs. We had previously noticed that *Xenopus* development is characterized by stage-specific fluctuations in steady state abundance of histone modifications, which we suspected to encode epigenetic information for the regulation of developmental processes [20]. These assumptions have been tested here by employing a small molecule inhibitor strategy (HUA), which had been pioneered by Harris and Hartenstein (1991). We have achieved a nearly complete arrest of cell proliferation from the late gastrula stage on, which we have used to monitor the consequences for both the histone modification landscape and for embryonic differentiation by RNA in situ hybridization for marker genes, while embryos develop into tadpoles.

Importantly, HUA treatment is not tolerated by younger (blastula) embryos, which die within a few hours after inhibitor application. Therefore, we cannot investigate with this method the immediate seeding of histone marks in the wake of zygotic genome activation at MZT. This temporal restriction provides a plausible explanation for how HUA-arrested embryos can finish the formation of germ layers and embryonic axes with the same efficiency as controls (Fig 3 and S1 Fig)—i.e., the cell cycle arrest occurs mainly after these events. With notable exceptions discussed below, this is reflected in the maintenance of many gene expression domains in HUA embryos (Fig 3).

While our analysis of the cellular composition and differentiated status of cells and organs is naturally limited, we note that HUA embryos at the tadpole stage are capable of muscle contraction, both spontaneously and in response to touch stimulation (S3 and S4 Videos). Furthermore, their epidermis contains multiciliated cells (see Fig 3C, alpha tubulin staining), which are fully differentiated at the tailbud stage. Finally, the HUAwo experiment demonstrated that morphological phenotypes, aberrant gene expression patterns, and the histone modification landscape are all largely recovering toward normality. We therefore conclude that changes in the histone modification landscape mainly arise as a consequence of the G1/S phase cell cycle arrest.

For an exploratory study like this, the limited replication negatively impacts the statistical interpretation of low abundant PTMs exerting high variance, such as H3S10Ph (see Figs 2 and 4B, S5 Table, and S1 and S6 Data), despite the large mean fold-change. On the contrary, for highly abundant modifications with very low variance, such as unmethylated H3 lysine 4 (H3K4p), much smaller effects appear statistically significant, while being most likely biologically irrelevant (Fig 4B and S5 Table). Therefore, the statistical results for any individual modification need to be interpreted with caution. While keeping this in mind, we focus our further discussion on the magnitude of the observed fold-changes for biologically relevant PTMs involved in gene regulation. We hope that the observed trends will stimulate independent experiments in other systems.

By far, the largest differences in abundance were measured for the mitotic H3S10Ph mark, catalyzed by aurora kinases [61]. Throughout the entire time course, the abundance of this G2/M mark was 12-fold reduced in arrested embryos. Therefore, proliferating cells represent only a minor contamination in HUA chromatin. On the basis of bulk histones, mass spectrometry has pointed out several key PTMs, whose abundance depends on whether embryonic cells are proliferating or not. As a reference, point serve H3K4me2/me3 marks, located at transcriptional start sites of active genes. Their abundance is well maintained under HUA treatment, indicating that the number, although not necessarily the identity, of active and poised promoters remains constant between the experimental conditions. In contrast, acetylated H3K9 and H3K27 marks are 3 to 6-fold diminished in arrested embryos and recover upon inhibitor washout. In functional terms, acetylation protects K9 and K27 residues from becoming decorated with repressive methyl marks. Specifically, the reduction in H3K27ac suggests a problem with the activation of distal enhancers, which diversify cell-type specific transcriptional programs during differentiation [45]. At the same time, HUA chromatin contains about three-fold elevated levels for transcriptionally repressive H3K9me3, H3K27me3, and H4K20me3. To evaluate this increase properly, one has to consider two aspects. First, based on an experimentally determined lengthening of the cell cycle from about 5 hours at gastrulation to about 12 hours at tailbud stage [28], HUA arrested embryonic cells probably miss only three to four rounds of mitosis. Secondly, the normal abundance of the repressive PTMs lies below 5% (S4 and S6 Figs), which is sufficient to control all heterochromatic and silenced parts of the genome. Therefore, a near three-fold increase bears a significant regulatory potential, either by enlarging the size of repressed chromatin domains, or by increasing the modification density on sites decorated at threshold levels such as bivalent chromatin domains [62]. Under the assumption that losses of acetyl marks on H3K9 and H3K27 are locally coupled to a gain in methyl marks, this altered histone landscape could decrease transcription of many genes in HUA embryos, which would be needed at higher expression levels under normal conditions. This can be investigated in organoid cultures, which compared to embryos have a more homogenous cellular composition and thus are amenable to RNA- and ChIP-Seq profiling.

Which kind of mechanisms link the cell cycle to the histone modification landscape? Studies following the acquisition of PTMs on newly incorporated histone proteins have revealed that H3K27me2/3, H3K79me1/me2 and H4K20me2/3 reach higher levels in starved cells than in cycling cells. This suggests that their steady state depends on the amount of time spent in G1/G0 [46,63,64]. This rationale may also apply to cells transiently arresting in the G2-phase, as it occurs, for instance, during neurogenesis in frogs and flies [27,65]. Although the chromatin of these cells has not been profiled, they are expected to contain maximal levels of the repressive mark H4K20me1 on newly incorporated histones, due to the G2/M-phase specific activity of PR-Set7 [66]. An increase in cell cycle length has been shown to control the nuclear localization of MET2. This histone methyltransferase is responsible for the timely accumulation of H3K9me2, which restricts cellular plasticity during *Caenorhabditis elegans*

embryogenesis [67]. While these examples illustrate cell cycle phase-specific influences on histone modifying enzymes, other mechanisms directly influence the inheritance of histone PTMs during S phase. Most modifications appear to follow the simple paradigm that new histones are modified until they become identical to the old ones, typically in one cell cycle round. In contrast, the modifications H3K9me3, H3K27me3, and probably H4K20me3 (our work here) are propagated with continuous but much slower kinetics on both old and new histones to restore the parental density [46]. While the mechanisms for these kinetic differences have not been resolved, they suggest that cell proliferation is rate limiting for the maintenance of some histone PTMs. Data from the adult mouse gut strongly support this notion. Intestinal stem cells with a low proliferative rate contain higher levels of H3K27me3 than their more proliferative descendants, i.e., transitory amplifying cells [54]. Furthermore, the conditional knockout of the PRC2 subunit EED in the intestine reactivates cell proliferation and causes derepression of genes proportional to the number of preceding cell divisions. Although most of these data rest on indirect evidence, these suggest a threshold level for H3K27me3, below which transcriptional repression is lost. These findings led the authors to conclude that replicational dilution is the major cause of H3K27me3 removal in mammalian cells [54]. Such a simple model may not be generally applicable to all modifications. Not only there are distinct modes of propagation (fast and slow), but also the combination of different histone marks matter. In the so-called domain model of histone propagation, preexisting methyl marks on K27 and K36 of the histone H3 tail mutually antagonize their respective methylation rates, such that the site of incorporation of new histones determines their ultimate methylation state. This mechanism is thought to enhance the stability of epigenetic states [60]. Based on such findings and our data presented here, we expect a highly diversified impact of the cell cycle on the epigenetic landscape.

Finally, does the altered chromatin landscape affect embryonic development? Our analysis allows only to draw indirect conclusions, since we cannot determine the individual distribution of histone modifications in the different experimental conditions due the cellular heterogeneity of the embryo. However, on the way from the tailbud stage to the phylotypic stage (NF28 to NF31), when the vertebrate body plan becomes apparent, we have noticed several differences between control and HUA embryos, which deserve to be discussed.

While gene expression occurs generally in synchrony between the 2 conditions (S1 Fig), HUA embryos fail to differentiate several externally visible structures. These include (i) melanocytes, i.e., descendants of the cranial neural crest, unambiguously identified by melanin production; (ii) a translucent fin, which forms on the dorsal midline along the main body axis; and (iii) a postanal tail. The absence of cell proliferation does not explain these phenotypes per se, since related organs are formed, as indicated by the small cluster of rx1-positive cells that is visible in most HUA embryos (Fig 3C) and defines the eye cup primordium. This suggests that either the inducing or the responding cells are compromised in HUA-arrested embryos. The up-regulation of repressive histone marks coupled with the loss in H3K9ac/H3K27ac marks would fit a scenario, in which genes required in the retina have not become activated to sufficient levels. A similar case can be made for the tail, which grows out from 2 spots of mesendoderm cells, located left and right of Spemann's organizer [68]. Cells at the tip of the tail maintain expression of the transcription factor genes *brachyury* (*xbra*) and *foxd5a*. Permanently arrested (HUA) embryos have lost the expression of these genes in the tail primordium, unlike transiently arrested (HUAwo) embryos, who maintain their expression and form a proper tail (Figs 3 and 5 and S1 Table). The tail represents a paradigmatic example, where gene expression could be compromised by changes in histone PTMs in response to cell cycle arrest. These hypotheses can be addressed and deserve further experimentation in the future.

In summary, we propose that, in vertebrate embryos, the duration of proliferation in cell lineages, and the frequency of mitosis, as a function of cell cycle length, can control the abundance of histone modifications via replicational dilution. As we have argued here, although not proven, this might be sufficient to control developmental decisions. In principle, PTMs with slow biological rate constants are predicted to be preferentially sensitive to this mechanism. On the other hand, cell populations may experience a particularly strong dilution effect, when they switch between resting and proliferating states. While this cell behavior occurs frequently in development [28,69–71], a universal impact for replicational dilution on the epigenetic landscape is expected in progenitor and stem cell populations.

## Materials and methods

### Ethics statement

Protocols of *Xenopus* care and experimental use are approved by the Government of Oberbayern (approval number: ROB-55.2-2532.Vet_03-17-102).

### Embryos handling and HUA treatment

*X. laevis* eggs were collected, in vitro fertilized, and handled as described in "Early Development of *Xenopus Laevis*: A Laboratory Manual" [72]. Embryos were staged accordingly to Nieukoop and Faber [73]. When they reached the desired stage, the embryos were immersed into a solution with 2 DNA synthesis inhibitors (HUA): 20 mM hydroxyurea and 150 µM aphidicolin (made from a frozen stock at 10 mg/ml in DMSO) in 0.1x MBS solution. Hydroxyurea blocks ribonucleotide diphosphate reductase, an enzyme that catalyzes the reductive conversion of ribonucleotides to deoxyribonucleotides, a crucial step in the biosynthesis of DNA. Aphidicolin blocks eukaryotic DNA Pol-alpha. To establish a full and irreversible cell cycle block, HUA treatment takes 4 to 6 hours, if kept in the solution continuously [33]. Within this time period, embryos develop from NF10.5 to NF13, if cultivated at 16C. Controls in all cases were embryos from the same batch that were treated identically, except they were pipetted into 2% DMSO in 0.1x MBS solution (Mock). HUA- and Mock-treated embryos were kept continuously in the HUA and Mock solutions, respectively, while in the HUAwo condition, the embryos were transiently incubated in the HUA from NF10.5 until NF13, and then moved in the Mock until the end point of the analysis, NF32.

### RNA in situ hybridization and immunocytochemistry

Whole-mount RNA in situ hybridization was performed as described in "Early Development of *Xenopus Laevis*: A Laboratory Manual" [72]. For immunocytochemistry, anti-H3S10Ph antibody (1:500, Active Motif, Carlsbad, CA, USA) and anti-mouse alkaline phosphatase–conjugated (1:2,000, Chemicon/Thermo Fisher Scientific, Waltham, MA, USA) secondary antibody were used. Embryos were photographed with a Leica M205FA stereomicroscope. Signal from H3S10Ph+ cells was counted using ImageJ software. For confocal imaging, anti-H3S10Ph antibody (1:500, Active Motif), anti-x-β-catenin antibody (1:100, Elizabeth Kremmer), and DAPI were used. Embryos were photographed with a Leica TCS SP5II confocal microscope, using Z-stack montage function. Z-stack depth, 30 µm.

### Nuclear histone extraction

Around 50 to 200 embryos developed to desired stages (NF13, NF18, NF25 and NF32) were harvested and washed with 110 mM KCl, 50 mM Tris/HCl (pH 7.4 at 23 uC), 5 mM MgCl2, 0.1 mM spermine, 0.1 mM EDTA, 2 mM DTT, 0.4 mM PMSF, and 10 mM Na-butyrate. Nuclei of

the embryos were prepared by centrifugation with 2,600x*g*, 10 minutes (3 to 18, Sigma) after homogenization by a 5-ml glass–glass douncer (Braun, Melsungen). The nuclear pellets of *Xenopus* embryos were resuspended in 1 ml 0.4 M HCl, incubated on a rotating wheel overnight and dialysed against 3 l of 0.1 M acetic acid/1 mM DTT. The dialysed histone solution was vacuum dried in a Concentrator Plus (Eppendorf) and stored at −20˚C. Each developmental stage in experiment type A is represented by 3 biological replicates; one biological replicate for experiment type B. Each biological replicate derives from a different mating pair.

## Histone acid extraction and sample preparation

The pellet from the nuclear histone extraction was dissolved in an appropriate amount of Laemmli Buffer to reach 1.37e6 nuclei/µl in each sample. A volume of 15 µL were loaded on an 8% to 16% gradient SDS-PAGE gel (SERVA Lot V140115-1) and stained with Coomassie Blue to visualize the histone bands. Histone bands were excised and propionylated as described before [74]. Propionylation blocks all endogenously unmodified and monomethylated lysine residues from being cleaved in the subsequent trypsin digest, thereby creating an optimized peptide pool for mass spectrometry analysis. Due to this step, naturally unmodified lysine residues are labeled as "p," tryptic peptides which have no modification states indicated as "un." Arginine10-labeled forms of the PTMs of interest (500 fmols/peptide, JPT Peptide Technologies, Berlin, Germany) were added to the gel samples together with trypsin. Digested peptides were sequentially desalted using C18 Stagetips (3M Empore, St Paul, MN, USA) and porous carbon material (TopTip Carbon TT2CAR (Glygen Corporation, Columbia, MD, USA)) as described elsewhere [75] and resuspended in 15 µl of 0.1% FA.

## Mass spectrometry analysis with scheduled PRM method

To study histone PTMs during *Xenopus* development, we analyzed the absolute and relative abundance of histones from HUA treated and control cohorts of embryos from the 4 developmental stages using LC–MS (S7 Fig).

As an internal and intersample control, a library consisting of isotopically labeled peptides (R10s) was used (S2 Table) to normalize for ionization differences between peptides. The R10 peptides were mixed in the library at equimolar concentration, and the mix was added to each analyzed sample. In total, the library consists of 64 peptides, 58 of them represent different histone H3 modification states, 6—histone H4 modification states. Considering only confirmed methylation and acetylation modifications on H3 and H4 histone tails, the library covers 76% of the histone H3 and 50% of the histone H4 modification states.

In order to identify and measure the abundance of the histone PTMs, we used a parallel reaction monitoring (PRM) method [76]. This method allows to isolate a target peptide ion based on the mass and retention time window, fragment it, and analyze the masses of all fragment ions simultaneously. Therefore, we could distinguish and identify peaks of isobaric peptides.

Peptide mixtures (5 µL) were subjected to nanoRP-LC-MS/MS analysis on an Ultimate 3000 nano chromatography system (Thermo Fisher Scientific, Dreieich Germany) coupled to a QExactive HF mass spectrometer (Thermo Fisher Scientific). The samples were directly injected in 0.1% formic acid into the separating column (150 × 0.075 mm, in house packed with ReprosilAQ-C18, Dr. Maisch GmbH, 2.4 µm) at a flow rate of 300 nL/min. The peptides were separated by a linear gradient from 3% ACN to 40% ACN in 90 minutes. The outlet of the column served as electrospray ionization emitter to transfer the peptide ions directly into the mass spectrometer. The QExactive HF was operated in a scheduled PRM mode to identify and quantify specific fragment ions of N-terminal peptides histone proteins. In this mode, the mass spectrometer automatically switches between one survey scan and 9 MS/MS acquisitions

of the m/z values described in the inclusion list containing the precursor ions, modifications, and fragmentation conditions (S3 Table). Survey full scan MS spectra (from m/z 270 to 730) were acquired with resolution 60,000 at m/z 400 (AGC target of $3 \times 10^6$). PRM spectra were acquired with resolution 30,000 to a target value of $2 \times 10^5$, maximum IT 60 ms, and isolation window 0.7 m/z and fragmented at 27% or 30% normalized collision energy. Typical mass spectrometric conditions were spray voltage, 1.5 kV; no sheath and auxiliary gas flow; and heated capillary temperature, 250˚C.

The mass spectrometry proteomics data have been deposited to the ProteomeXchange Consortium via the PRIDE partner repository [77] with the data set identifier PXD021803.

## Histone PTM quantification

Data analysis was performed with the Skyline (version 3.7) [78] by using doubly and triply charged peptide masses for extracted ion chromatograms (XICs). Selection of respective peaks was identified based on the retention time and fragmentation spectra of the spiked in heavy-labeled peptides. Integrated peak values (total area MS1) were exported as.csv file for further calculations. Total area MS1 from endogenous peptides was normalized to the respective area of heavy-labeled peptides. The sum of all normalized total area MS1 values of the same isotopically modified peptide in one sample resembled the amount of total peptide. The relative abundance of an observed modified peptide was calculated as percentage of the overall peptide.

## RNA extraction and qPCR sample preparation

Total RNA of 10 embryos was extracted using Trizol (Ambion/Thermo Fisher Scientific, Dreieich, Germany) and phenol/chloroform. The RNA was precipitated with 70% isopropanol and cleaned using the RNeasy Cleanup Kit (Qiagen GmbH, Quiagen Strasse 1, D-40724 Hilden Germany) including DNAse I-on-column digestion. For qPCR analysis, 1 μg of total RNA was transcribed with the DyNAmo cDNA Synthesis Kit (Thermo Fisher Scientific). For qPCR, 5 to 20 ng cDNA was mixed with the Fast SYBR Green Master mix (Applied Biosystems/Thermo Fisher Scientific, Dreieich, Germany) and amplified with a LightCycler (Roche Diagnostics, Mannheim, Germany). Primer sequences are given in S4 Table.

## Statistical analysis and data visualization

For embryonic quantitative analysis (morphological phenotype, qRT/PCR, quantitative MS analysis), SEMs are displayed. Statistical analysis was performed using two-tailed, paired Student *t* test. In case of multiple testing, the Benjamini–Hochberg FDR procedure was performed—indicated as "adj.p." For boxplots in Figs 2B and 5A, "ggplot2" R package was used. PCA in S3A Fig was performed using R, without scaling. MA-plot, volcano plot, and boxplot in S3B–S3D Fig were generated using GraphPad Prism v9.0.1.

## Heatmap generation

Mass spectrometry intensity values of the spiketides were log2 transformed, and missing values imputed using 2 nearest neighbors (library "knnImputation" in R). Endogenous peptides intensities were log2 transformed after adding 1 to all values. For heatmap display in Fig 4A, the normalized log ratios were quantile normalized across all samples and subsequently averaged per condition (*n* = 3 biological replicates per stage per condition). The values were scaled row-wise per peptide and hierarchically clustered using the "complete" method on euclidean distances. For heatmap generation and hierarchical clustering in Fig 6A, values of relative histone PTM abundance were used (*n* = 1 biological replicate at developmental stage NF32).

## Supporting information

**S1 Fig. HUA treatment reduces survival and impacts morphogenesis.** (A) Embryonic survival curves under Mock, HUA, or HUAwo condition. Data from n $\geq$ 3 biological replicates/condition; mean ± SEM. The individual quantitative observations can be found in S5 Data (sheet 1). (B) The first morphological effect of HUA treatment is apparent at stage NF19 as a delay in neural tube closure (in mock, black arrowhead points to the dorsal midline; in HUA, 2 black arrows point to separate neural folds). Under higher magnification, HUA embryos contain larger cells. After hatching (stage NF37/38), HUA-treated embryos lack tails, have reduced eyes and malformed fins, and are largely deficient in melanocytes. (C) Comparison of temporal expression profiles for selected marker genes in Mock and HUA conditions by qRT/PCR, normalized to *odc* mRNA. Genes are grouped according to their activation time point. *N* = 3 biological replicates/condition; mean ± SEM. Significant difference was detected only in case of *pax6* expression level at NF32 stage (Student *t* test [two-tailed, paired]; * $p < 0.05$). No other significant differences were detected between the 2 conditions. The individual quantitative observations can be found in S5 Data (sheet 2). CNS, central nervous system; HUA, hydroxyurea and aphidicolin; HUAwo, HUA washout; qRT/PCR, quantitative reverse transcription polymerase chain reaction.
(TIF)

**S2 Fig. Apoptosis in HUA- and Mock-treated embryos.** Wild-type embryos were injected with FADD apoptosis inducing plasmid in one blastomere at 4-cell stage as a positive control. FADD-injected and FADD-uninjected wild-type embryos together with Mock and HUA embryos were ICC stained against activated cas-3. Signal from cas-3 staining can be observed on the skin of the embryos as small blue dots; dashed ovals highlight FADD-induced cas-3 staining; white asterisks indicate blastocoel background staining, scale bar: 1 mm. HUA embryos do not demonstrate an increased level of apoptosis compared to Mock siblings. cas-3, caspase-3; FADD, Fas-associated protein with death domain; HUA, hydroxyurea and aphidicolin; ICC, immunocytochemical staining.
(TIF)

**S3 Fig. Visualization of the data from experimental series A and B.** (A) PCA for Mock- and HUA-treated HPMs (Exp. A). Each data point represents 64 modification states, measured by LC–MS/MS in PRM mode, with absolute abundance calculated with R10 spiketide normalization. Mock and HUA data sets are partially separated, with younger HUA samples intermingling with older Mock samples. (B) MA-plot detailing the distribution of Log2 transformed relative histone PTM abundance [Log2(%)] over Log2 transformed fold-change between HUA and Mock conditions [Log2(fold-change)] (Exp. A). Histone PTMs different between HUA and Mock conditions with adj.*p* < 0.1 (Benjamini–Hochberg procedure) highlighted in red. Lower abundant modifications tend to have larger variability. (C) Volcano plot shows the distribution of Log2 transformed fold-change [Log2(fold-change)] over negative Log2 transformed adj.*p*-value [−Log2(adj.*p*)] between HUA and Mock conditions (Exp. A). Baseline in gray: 3.3219 as −Log2(0.1). Histone PTMs different between HUA and Mock with adj.*p* < 0.1 (Benjamini–Hochberg procedure) are above the baseline, highlighted in red. One-quarter of the analyzed histone PTMs is significantly different between Mock and HUA conditions. (D) Boxplot indicates the distribution of Log2 transformed fold-changes [Log2(fold-change)] between HUA and Mock conditions in Exp. A and Exp. B. Median is shown. The results from Exp. A and Exp. B are similar. The direct comparison of relative abundance of the histone PTMs is shown in S6 Fig. The individual quantitative observations can be found in S6 Data. HMP, histone modification profile; HUA, hydroxyurea and aphidicolin; LC–MS/MS, liquid

chromatography–tandem mass spectrometry; PCA, principal component analysis; PRM, parallel reaction monitoring; PTM, posttranslational modification of histone.
(TIF)

**S4 Fig. Relative histone PTM abundance in HUA- and Mock-treated embryos.** (A) Individual relative histone PTM distribution for histone H3. Data are first normalized to R10 spiketide signals, then added up to 100% for all modification states measured for each specific tryptic peptide, from which the relative contribution of each state is then calculated. The insert in H3 9–17 K9/S10/K14 plot shows a zoom-in for the H3S10Ph mark. (B) Individual relative histone PTM distribution for histone H4. *N* = 3 biological replicates/condition; mean ± SEM. "p," propionylated (naturally unmodified). The individual quantitative observations can be found in S5 Table. ac, acetylation; HUA, hydroxyurea and aphidicolin; me, methylation; Ph, phosphorylation; PTM, posttranslational modification of histone.
(TIF)

**S5 Fig. Recovery of mitotic activity in HUAwo embryos.** ICC for the mitotic histone mark H3S10Ph at indicated stages. Mitotic cells are marked by black dots. Elongated, older embryos are recorded as anterior halves, i.e., at the same magnification as younger stages, and in whole mount views as inserts. Scale bars: 1 mm. N ≥ 3 biological replicates/condition. HUAwo, HUA washout; ICC, immunocytochemical staining.
(TIF)

**S6 Fig. Comparison of relative histone PTM abundance from experimental series type A and B.** Relative histone PTMs abundance was calculated as described in Materials and methods. The insert in H3 9–17 K9/S10/K14 plot shows a zoom-in for the H3S10Ph mark. (A) Color coding of sample types. Panels (B) and (C): individual relative histone PTM distributions for histone H3 and H4, respectively. "p," propionylated (naturally unmodified). The individual quantitative observations can be found in S6 Table. ac, acetylation; me, methylation; Ph, phosphorylation; PTM, posttranslational modification of histone.
(TIF)

**S7 Fig. Absolute quantification of histone posttranslational modification states by LC–MS/MS using scheduled PRM method.** (A) Pipeline of mass spectrometry analysis of histone modifications from *Xenopus laevis*. Bulk histones are isolated from purified nuclei of embryos from 4 sampled stages (see Fig 1) by acidic extraction and SDS-PAGE. Propionylation blocks all endogenously unmodified and monomethylated lysine residues from being cleaved in the subsequent trypsin digest, thereby creating an optimized peptide pool for Mass Spec analysis. Due to this step, naturally unmodified lysine residues are labeled as "p," tryptic peptides which have no modification states indicated as "un." After propionylation, but before trypsin digest, we add to each sample a so-called R10 library (S2 Table), which consists of isotopically heavy-labeled arginine peptides (R10). The individual R10 peptides are mixed in equimolar concentration and mimic 64 histone H3 and H4 modification states. These isotopically heavy-labeled peptides serve as an internal and intersample control, allowing to minimize technical variations and to quantitate abundance of histone modification states on the absolute scale. (B) Representation of the R10 spike-in peptide control. Each of the analyzed endogenous histone modification states has a synthetized R10 peptide analog. Due to the same chemical properties, endogenous tryptic peptides and their R10 spiketide analogs elute at the same RT; however, they can be distinguished based on the mass to charge (m/z) ratio. Additionally, R10 spiketides help with peak identification based on RT and detail fragmentation spectra for isobaric peptides. LC–MS/MS, liquid chromatography–tandem mass spectrometry; PRM, parallel reaction

monitoring; RT, retention time.
(TIF)

**S1 Table. A panel with whole mount in situ gene markers analysis.**
(XLSX)

**S2 Table. A list of isotopically heavy-labeled peptides (R10 spiketides).**
(XLSX)

**S3 Table. An inclusion list for scheduled PRM MS analysis.**
(XLSX)

**S4 Table. A list of qRT/PCR primers.**
(XLSX)

**S5 Table. Results from the histone PTM MS measurements from Exp. A.**
(XLSX)

**S6 Table. Results from the histone PTM MS measurements from Exp. B.**
(XLSX)

**S1 Video. Z-stack of the Mock NF25 embryo.**
(AVI)

**S2 Video. Z-stack of the HUA NF25 embryo.**
(AVI)

**S3 Video. Flight response of Mock NF33 embryos.**
(MP4)

**S4 Video. Flight response of HUA NF33 embryos.**
(MP4)

**S1 Data. Raw numerical values of the H3S10Ph-positive cells used for the boxplot (Fig 2B).**
(XLSX)

**S2 Data. Raw numerical values of the endogenous (Sheet 1 "endogenous") and R10 spiketides (Sheet 2 "Spike-in R10") histone PTM abundance, as derived from the MS measurements.** R10 normalized values used for the absolute heatmap (Fig 4A) are in the Sheet 3 ("normalized").
(XLSX)

**S3 Data. Raw numerical values of the H3S10Ph-positive cells used for the boxplot (Fig 5A).**
(XLSX)

**S4 Data. Raw numerical values of the endogenous (Sheet 1 "endogenous") and R10 spiketides (Sheet 2 "Spike-in R10") histone PTM abundance, as derived from the MS measurements.** Values of the relative abundance of the indicated histone modification states, which were used for the heatmap (Fig 6A), are in Sheet 3 ("%"). Borders separate histone modification states within one tryptic peptide.
(XLSX)

**S5 Data. Sheet 1 "S1A survival"–raw numerical values used for the embryonic survival curves (S1A Fig).** When not available (NA), embryos were taken for other experimentation or the time point was not assessed. Sheet 2 "S1C RT-qPCR"–raw numerical values used for the

RT-qPCR (S1C Fig).
(XLSX)

**S6 Data. Sheet 1 "S3A PCA"–numerical values used for the PCA (S3A Fig).** They are the same R10 normalized values as in Fig 4A Data, Sheet 3 ("normalized"). Sheet 2 "S3B MA-plot"–calculated Log2[%] and Log2[fold-change] values used for the MA-plot (S3B Fig). Sheet 3 "S3C"–calculated Log2[fold-change] and -Log2[adj.*p*] values used for the Volcano-plot (S3C Fig). Sheet 4 "S3D ratio boxplot"–calculated Log2 of fold-change HUA/Mock in experiment A and B values used for the ratio boxplot (S3D Fig).
(XLSX)

## Acknowledgments

We express our gratitude to Edith Mentele and Barbara Hölscher for their expert technical support in RNA in situ hybridization and immunocytochemical staining. We thank Lea Schuh for her comments.

## Author Contributions

**Conceptualization:** Daniil Pokrovsky, Ralph A. W. Rupp.

**Data curation:** Daniil Pokrovsky, Ignasi Forné.

**Formal analysis:** Daniil Pokrovsky, Tobias Straub.

**Funding acquisition:** Ralph A. W. Rupp.

**Investigation:** Daniil Pokrovsky.

**Methodology:** Daniil Pokrovsky, Ignasi Forné.

**Project administration:** Daniil Pokrovsky, Ignasi Forné, Axel Imhof, Ralph A. W. Rupp.

**Resources:** Axel Imhof, Ralph A. W. Rupp.

**Software:** Tobias Straub.

**Supervision:** Ignasi Forné, Axel Imhof, Ralph A. W. Rupp.

**Validation:** Daniil Pokrovsky.

**Visualization:** Daniil Pokrovsky, Tobias Straub.

**Writing – original draft:** Daniil Pokrovsky.

**Writing – review & editing:** Daniil Pokrovsky, Ignasi Forné, Tobias Straub, Axel Imhof, Ralph A. W. Rupp.

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
