## [Editor Report · Decision Letter 0]

1 Sep 2020

Dear Dr Rupp, 

Thank you for submitting your manuscript entitled "Mitotic activity shapes stage-specific histone modification profiles during Xenopus embryogenesis" for consideration as a Research Article by PLOS Biology.

Your manuscript has now been evaluated by the PLOS Biology editorial staff as well as by an academic editor with relevant expertise and I am writing to let you know that we would like to send your submission out for external peer review.

Please re-submit your manuscript within two working days, i.e. by Sep 03 2020 11:59PM.

Kind regards,

Ines

--

Ines Alvarez-Garcia, PhD

Senior Editor

PLOS Biology

---

## [Decision Letter · Decision Letter 1]

10 Dec 2020

Dear Dr Rupp,

Thank you very much for submitting your manuscript entitled "Mitotic activity shapes stage-specific histone modification profiles during Xenopus embryogenesis" for consideration as a Research Article at PLOS Biology. Thank you also for your patience as we completed our editorial process, and please accept my sincere apologies for the delay in providing you with our decision. Your manuscript has been evaluated by the PLOS Biology editors, an Academic Editor with relevant expertise, and we consulted three independent reviewers, although we have to date only received reports from two of them; we will forward you the third one if it is sent to us belatedly.

As you will see, both reviewers find the results and conclusions interesting, but they also raise several points and concerns that need to be addressed before we can consider the manuscript further. The reviewers ask you to consider and test potential alternative interpretations of the results, and they also point out that some of the data have not been analysed in a statistically robust manner. In addition, they note several points that need to be clarified.

In light of the reviews (attached below), we will not be able to accept the current version of the manuscript, but we would welcome re-submission of a much-revised version that addresses all the reviewers' comments. We cannot make any decision about publication until we have seen the revised manuscript and your response to the reviewers' comments. Your revised manuscript is also likely to be sent for further evaluation by the reviewers.

We expect to receive your revised manuscript within 3 months. 

**IMPORTANT - SUBMITTING YOUR REVISION**

*Re-submission Checklist*

*Published Peer Review*

*PLOS Data Policy*

*Blot and Gel Data Policy*

Sincerely,

Ines

--

Ines Alvarez-Garcia, PhD,

Senior Editor,

PLOS Biology

Reviewers’ comments

Rev. 1:

This is an interesting manuscript that investigates the relationship between chromatin modifications and proliferation in the Xenopus embryos from gastrula to tailbud stages. They inhibit DNA replication with a combination of hydroxyurea and aphidicolin and then assess morphology and bulk chromatin modifications by mass spec. Consistent with previous work from others, embryos undergo mostly normal morphogenesis through early tailbud, aside from some differences in the size of eye and tail precursors and a relative enlargement of individual cell sizes. The most novel and interesting aspect of the work is the definition of a "stage-specific histone modification profile" (HMP) for each stage of development; these HMPs appear to be distinct from one another and are a novel method of characterizing global histone modifications in development. The authors show that exposure to HUA dramatically alters the HMPs at each stage and that this can be reversed by washing out the HUA. Overall, this is an interesting and thoughtful body of work, and I do not have experimental concerns, but I do have suggestions for clarification or revision to the text.

1. The authors at times assume causality when this is not supported by the data. For example, the statement "These results indicate that replicational dilution of histone marks has a strong impact on developmental chromatin maturation" is not yet supported by the data. They have not shown that proliferation or cell cycle progression directly affects chromatin state. They show that the HMP varies with stage and that replication inhibitors alter these profiles, but it is not yet clear that the changes caused by exposure to HUA are due to inhibition of proliferation or to some other effect of these drugs. This could include replication stress and DNA damage, for example (see point #2). Similarly, how do they know that "Mitotic activity shapes stage-specific histone modification profiles"? Again, this could be an effect of either drug that is independent of cell cycle. There are other instances throughout the manuscript. This concern can be addressed by modifying the text.

2. The authors should address (in the text) how they can distinguish whether the altered chromatin modification profile after HUA treatment is due to cell cycle arrest rather than a response to replication stress or DNA damage, which are associated with HU treatment. If they cannot rule these out, then they should at least discuss this limitation.

3. Very similar work addressing the effect of aphidicolin on morphogenesis was published by Rollins and Andrews in Development [PMID 1794324], and they came to similar conclusions about the impact of proliferation on differentiation during neurula and tailbud stages. This work should be discussed and cited in the introduction along with Harris and Hartenstein. The current manuscript makes a substantial advance beyond the 1991 Rollins paper, but it is an important and relevant background paper.

4. Could the authors explain why they used both hydroxyurea and aphidicolin? Wouldn't aphidicolin be sufficient to inhibit proliferation, as in the Rollins paper?

5. Fig 4B shows change in histone PTMs in HUA relative to control. I did not see an indication of how or whether they assessed the statistical significance of these changes. This should be included and mentioned either in the results section or the figure legend.

Minor:

6. On the first page of Results, it is not clear what is meant by "lies behind" in the sentence "The end point of the analysis (NF32) lies behind the phylotypic stage of Xenopus (NF28-31), when evolutionary conserved gene expression programs have established the vertebrate body plan." Do they mean the "end point" is after the phylotypic stage?

7. The control to rule out developmental delay is important; they might consider moving to main figures. That's just a suggestion.

Overall, this is interesting and important work.

Rev. 2:

Pokrovsky et al. use mass spectrometry to analyse multiple histone modifications in developing Xenopus embryos and investigate the effect of permanently or temporarily withdrawing the cells from the cell cycle on changes in histone modification states. The study tackles a very interesting question: the extent to which passage through DNA replication and mitosis influences the histone modification status in cells, particularly during development. This is important because histone PTMs are thought to play an important role in generating and/or maintaining the gene expression program of differentiating cells. The paper is clearly written and the basic cell cycle and developmental parameters of the model are outlined well. The study potentially provides some useful data but is rather preliminary in its current form. It is not clear if the effects of hydroxyurea and aphidicolin can be ascribed only to withdrawal from the cell cycle. Effects such as replication stress and DNA damage responses, and also marked changes in cell size, could also be involved. Finally, it is not clear that the results have been analyzed in a statistically robust manner.

Major points

1. In Figures 4 and 6, the core results are provided as Z-scores. However, it is clear that all the scores are fairly low, with few exceeding +/-2, and are therefore are not very far from the center of the distribution. Figures S4 and S6 show the variability of the raw results, but it would be very useful to have some sense of the variability and statistical significance of the derived results (Z-scores and ratios) in the main Figures 4 and 6.

2. Related to point 1, the absolute amount of H3S10ph peptides quantified appears very low (Fig S5 and S6). How does this affect the robustness of the findings? Are the results for low abundance marks generally less reliable than others? It is notable that another well-described mitotic marker, H3T3ph, shows the opposite pattern and apparently increases in HUA conditions. How can this be explained?

3. Also related to the above two points, how were the specific marks shown in Figures 4B and 6B selected? In the main, these are the most frequently discussed marks in the literature (eg H3K4me, H3K9me, H3K9ac, H3K27me), but I wonder if these are also the statistically most robust changes? Some notable marks such as H3K79me are ignored. Comparing the results of Mock B in Figure 6A with Mock A in Figure 4A there seem to be modifications which do not show such consistent changes in experiments A and B, though this is difficult to evaluate by eye.

4. It is unclear how similar the clustering is in Figures 4A and 6A. If the results are robust, presumably these might be expected to be somewhat similar?

5. In Figure S1, I think there is confusion here between the inability to see a "statistically significant" difference between two results and the conclusion that this means there is no difference. For example, if I measure the height of 3 men and 3 women, I may very well conclude that there is no statistically significant difference. This does not mean that there is no difference in the average height of men and women. It simply means that the natural variation in height (coupled with measurement error) is too large for me to figure this out with a small sample size. What level of difference could have been detected with this assay given the variation observed? This is important so that we can evaluate the evidence that gene expression does not change.

6. I am not convinced that the HUA washout experiment allows us to "distinguish" "non-physiological derailment" from "the consequence of the G1/S arrest", and to "conclude that changes in the histone modification landscape arise as a consequence of the G1/S-phase cell cycle arrest". This experiment is certainly useful, but perhaps it could be introduced and interpreted more cautiously. It is quite possible that HUA causes non-physiological effects such as DNA damage and increased cell size, but that these are reversible. Confirming the results with an "orthogonal" approach to withdraw cells from the cell cycle would be very valuable.

Minor points

1. There is a standard nomenclature for histone modifications, and it is not clear why there are inconsistent deviations from this in the paper. For example, methylation is sometimes "me" (the standard) and sometimes "m", sometimes "Me", and even occasionally "met" (Table S2). Standard notation for phosphorylation is "ph", but is typically "Ph" in this paper, and even "p" in Table S2. This latter usage is particularly confusing because "p" is used throughout the other figures and tables in a way that is not defined, but is evidently not for phosphorylation. Does this "p" refer to propionylation? If so, it should be prominently defined in the text.

2. Propionylation seems an important step in this process, but the methods do not seem to mention it.

3. Line colors do not match the key in Fig S1A.

4. In the Methods section on Heatmap Generation, it mentions n = 3, but this is only true for Figure 4A. For Figure 6A, n= 1.

---

## [Decision Letter · Decision Letter 2]

30 Apr 2021

Dear Dr Rupp,

Thank you very much for submitting a revised version of your manuscript entitled "A systemic cell cycle block impacts stage-specific histone modification profiles during Xenopus embryogenesis" for consideration as a Research Article at PLOS Biology. This revised version of your manuscript has been evaluated by the PLOS Biology editors, the Academic Editor and one of the original reviewers.

You will see that the reviewer appreciate the improvements you have done in the manuscript, but also has a remaining concern regarding the use of paired Student t tests in the statistical analysis of the data. The reviewer thinks you need to take into account the fact that you are making multiple comparisons. Considering this, you should convincingly show that the changes observed in repressive histone modifications are statistically significant.

In light of the review (attached below), we are pleased to offer you the opportunity to address the points raised by this reviewer in a revised version that we anticipate should not take you very long. We will then assess your revised manuscript and your response to the reviewers' comments and we may consult the reviewer again.

We expect to receive your revised manuscript within 1 month.

**IMPORTANT - SUBMITTING YOUR REVISION**

3. Resubmission Checklist

a) *Published Peer Review*

b) *PLOS Data Policy*

Please make sure you mention in the corresponding figure legends where the data can be found:

d) *Blurb*

Please also provide a blurb which (if accepted) will be included in our weekly and monthly Electronic Table of Contents, sent out to readers of PLOS Biology, and may be used to promote your article in social media. The blurb should be about 30-40 words long and is subject to editorial changes. It should, without exaggeration, entice people to read your manuscript. It should not be redundant with the title and should not contain acronyms or abbreviations. For examples, view our author guidelines: https://journals.plos.org/plosbiology/s/revising-your-manuscript#loc-blurb

Sincerely,

Ines

--

Ines Alvarez-Garcia, PhD

Senior Editor,

PLOS Biology

Reviewers' comments

Rev. 2:

The authors have done a very good job adjusting the text to reflect the points made by the reviewers. In particular, they have modulated the language to better reflect the results. I do have a substantial remaining concern however. Usefully, the authors have now provided some statistical analysis of their data. For this, they use paired Student t tests. However, no account is taken of the fact that they are making multiple comparisons. This is very important here, and I do not believe the manuscript should be published in its current form. For example, in Figure 4B, they make 88 comparisons (and actually many more if you take the whole dataset into account - see Supplemental Tables), but use unadjusted p values. Essentially, they have to take into account the principle that it more likely that you'll get a head on a coin if you throw it multiple times than if you throw it once. Whether publication is then warranted would depend, of course, on the extent to which they then convincingly show statistically significant changes in repressive histone modifications. The fact that the magnitude of the changes they observe are, as they admit, small, reinforces the importance of this analysis.

---

## [Decision Letter · Decision Letter 3]

2 Jul 2021

Dear Dr Rupp,

Thank you for submitting your revised Research Article entitled "A systemic cell cycle block impacts stage-specific histone modification profiles during Xenopus embryogenesis" for publication in PLOS Biology. Please accept again my apologies for the delay - we were unable to obtain advice from the previous Academic Editor and had to find a new one for the final decision. I have now received advice from Reviewer 2 and have discussed these comments with the new Academic Editor. 

Based on the reviews, we will probably accept this manuscript for publication, provided you satisfactorily address the last remaining point raised by Reviewer 2. As you will see, this reviewer is now mostly satisfied, but you will need to include a short discussion of the extent of fold-changes, especially the smaller but statistically significant ones, and their biological relevance as well as the uncertainty associated with some of these measurements. Please also make sure to address the all the data and other policy-related requests requested below.

We expect to receive your revised manuscript within two weeks. 

*Published Peer Review History*

*Early Version*

Sincerely,

Ines

--

Ines Alvarez-Garcia, PhD,

Senior Editor,

ialvarez-garcia@plos.org,

PLOS Biology

ETHICS STATEMENT:

-- Please include the full name of the IACUC/ethics committee that reviewed and approved the animal care and use protocol/permit/project license. Please also include an approval number.

Fig. 2B; Fig. 4A; Fig. 5A; Fig. 6A; Fig. S1A, C; Fig. S3A-D; Fig. S4A-B and Fig. S6A-B

Reviewers' comments

Rev. 2:

The authors have carried out multiple comparison adjustments as requested. The significance of most of the results fail to reach the typically used cut off of p < 0.05, which itself is not very stringent, and the authors instead use p < 0.1. This means that the relatively low fold-changes seen in this work are also not in general of huge statistical significance. Ideally, the paper should provide an explanation of why this atypical significance cut-off is used, together with a clear discussion of how the results should be interpreted. I think the validity of changes in any single histone modification in this analysis has to be considered very cautiously (for any given mark, the change seen could be observed in up to 1 in 10 experiments even on the null hypothesis), but the broader conclusion that some histone marks in general do change is probably reasonable.

---

## [Editor Report · Decision Letter 4]

30 Jul 2021

Dear Dr Rupp,

On behalf of my colleagues and the Academic Editor, Tom Misteli, I am pleased to say that we can in principle offer to publish your Research Article entitled "A systemic cell cycle block impacts stage-specific histone modification profiles during Xenopus embryogenesis" in PLOS Biology, provided you address any remaining formatting and reporting issues. These will be detailed in an email that will follow this letter and that you will usually receive within 2-3 business days, during which time no action is required from you. Please note that we will not be able to formally accept your manuscript and schedule it for publication until you have made the required changes.

PRESS

Sincerely, 

Ines

--

Ines Alvarez-Garcia, PhD 

Senior Editor 

PLOS Biology
